# Out-of-equilibrium dynamics of the Kitaev model on the Bethe lattice via coupled Heisenberg equations

Oleksandr Gamayun[1] and Oleg Lychkovskiy[2,3]⋆

**1** Faculty of Physics, University of Warsaw, ul. Pasteura 5, 02-093 Warsaw, Poland
**2** Skolkovo Institute of Science and Technology,
Bolshoy Boulevard 30, bld. 1, Moscow 121205, Russia
**3** Department of Mathematical Methods for Quantum Technologies,
Steklov Mathematical Institute of Russian Academy of Sciences
8 Gubkina St., Moscow 119991, Russia

⋆ o.lychkovskiy@skoltech.ru

## Abstract

The Kitaev model on the honeycomb lattice, while being integrable via the spin-fermion mapping, has generally resisted an analytical treatment of the far-from-equilibrium dynamics due to the extensive number of relevant configurations of conserved charges. Here we study a close proxy of this model, the isotropic Kitaev spin-1/2 model on the Bethe lattice. Instead of relying on the spin-fermion mapping, we take a straightforward approach of solving Heisenberg equations for a tailored subset of spin operators. The simplest operator in this subset corresponds to the energy contribution of a single bond direction. As an example, we calculate the time-dependent expectation value of this observable for a factorized translation-invariant (or staggered-translation-invariant) initial state with arbitrary initial (staggered) polarization.



# 1   Introduction

Integrability of a quantum many-body model does not automatically imply that its dynamics can be easily tracked. Quite the opposite, the dynamics of many models, whose spectrum and eigenstates have been found decades ago, is still under active investigation. Among techniques used in such studies are the quench action approach [1, 2], generalized hydrodynamics [3–5] and advanced machinery for summing form factor expansions [6–19]. Recently one of us has explored a straightforward approach to integrable dynamics based on solving an infinite set of coupled Heisenberg equations [20]. For a nonintegrable system this set is generally believed to be intractable without approximations, with the complexity of the involved operators growing unwieldy [21–23]. In contrast, for an integrable system one may hope to find a relatively small subset of relatively simple operators closed with respect to commutation with the Hamiltonian, and analytically solve the resulting system of linear differential equations. This approach has been successfully applied to integrable systems with Onsager algebra – the transverse-field Ising model and the superintegrable chiral Potts models [20].

Here we apply this approach to yet another model – the isotropic Kitaev spin-1/2 model on the Bethe lattice of degree 3. It is a close proxy to the Kitaev model on the honeycomb lattice [24]. Kitaev has demonstrated that his spin model with $N$ spins can be mapped to a fermionic model that becomes quadratic after fixing the values of $\sim N/2$ explicitly known integrals of motion (IoMs). In fact, an analogous model with a similar integrability structure can be defined on a large class of regular tricoordinate lattices (i.e. lattices with every vertex attached to 3 links) that can be divided in two sublattices, $A$ and $B$ [25–28]. In such lattices each link connects vertexes from different sublattices, and all links can be classified in three types – $X$-, $Y$- and $Z$-links. The Bethe lattice of degree 3 falls in this category, see Fig. 1. The Kitaev Hamiltonian reads

$$H = J_x \sum_{X-\text{links}} \sigma_{\mathbf{j}A}^x \sigma_{\mathbf{j}'B}^x + J_y \sum_{Y-\text{links}} \sigma_{\mathbf{j}A}^y \sigma_{\mathbf{j}'B}^y + J_z \sum_{Z-\text{links}} \sigma_{\mathbf{j}A}^z \sigma_{\mathbf{j}'B}^z, \tag{1}$$

where $\sigma_{\mathbf{j}A}^{x,y,z}$ and $\sigma_{\mathbf{j}'B}^{x,y,z}$ are Pauli matrices of the spins residing on sublattices $A$ and $B$, respectively, each link gives rise to a two-spin interaction term, and $J_x, J_y$ and $J_z$ are coupling constants. In our work, we will focus on the isotropic case with

$$J_x = J_y = J_z = -1. \tag{2}$$

An important difference between the honeycomb and Bethe lattices is that the latter lacks closed loops, in contrast to the former. The lack of loops will prove instrumental for obtaining a tractable system of equations.

When the Kitaev's spin-fermion mapping is employed, the complexity of describing the dynamics starting from a non-equilibrium initial state depends on the complexity of decomposition of this state in the joined eigenbasis of $\sim N/2$ IoMs fixed in the course of diagonalization: One has to solve a free-fermionic problem for each element of this decomposition separately.

In particular, if the initial state is an eigenstate of all these IoMs, a single free-fermionic problem should be solved, and analytical results can be obtained [29–33]. On the other hand, if the initial state is a superposition of an exponential number of these eigenstates (this is the case e.g. for product initial states), one faces a challenge of solving an exponential number of different disordered free-fermionic problems. Up to date, this challenge has been addressed only numerically by stochastic sampling the effective disorder produced by different configurations of values of IoMs [34, 35].

The approach we undertake in the present paper is completely different. It does not involve any spin-fermion mapping. Instead, we craft an infinite yet manageable subset of operators that is described by a closed system of Heisenberg equations. This construction is based on the algebra of *string operators* known to be closed with respect to the commutation [36, 37]. We solve the Heisenberg equations and obtain explicit expressions for each of the Heisenberg operators from the set. The simplest operator corresponds to the contribution to the total energy from a single bond direction. We calculate time-dependent expectation value of this operator for the initial translation-invariant or staggered-translation-invariant product state with an arbitrary (staggered) polarization.

The paper is organised as follows. In the next section we introduce the notion of strings and string operators. In Section 3 we derive a system of Heisenberg equations for a tailored set of operators, and in Section 4 these equations are solved. In Section 5 we calculate the time-dependent expectation value of the bond operator for a product initial state. We compare it to the analogous quantity calculated numerically for the honeycomb lattice [34] and find that they are quite similar, consistent with the physical intuition. The last section is devoted to the summary and outlook.

## 2 Strings on the Bethe lattice

### 2.1 Routes

We remind that a path on a graph is a sequence of links joining a sequence of vertices (of course, in the latter sequence any two consecutive vertices should be neighbouring). Bethe lattice supports only *self-avoiding* paths where all links and vertices are distinct.

A path on the Bethe lattice of degree three (or, in fact, on any graph with every vertex having degree 3) can be conveniently specified with the help of a *route* – a sequence of turns, left or right. The formal definition of a route and some related definitions are listed below.

- A *route* is a sequence containing two elements ("turns"), $l$ (left turn) and $r$ (right turn). We reserve calligraphic capital letters $\mathcal{V}$ and $\mathcal{W}$ for routes. As an example, we consider two specific routes,

$$\mathcal{V} = rll \quad \text{and} \quad \mathcal{W} = lr. \tag{3}$$

  These two routes will be used to exemplify various notions and definitions in what follows. In addition, we define the empty route $\emptyset$ containing no turns.

- $|\mathcal{V}|$ is the *length* (i.e. the number of turns) of the route $\mathcal{V}$. For the example (3)

$$|\mathcal{V}| = 3, \qquad |\mathcal{W}| = 2.$$

  By definition, $|\emptyset| = 0$.

- We define a function

$$\text{sign}\,\mathcal{V} = \pm 1.$$

It admits the value $-1$ when the route $\mathcal{V}$ contains an odd number of left turns, $l$, and $+1$ otherwise. For routes (3)

$$\operatorname{sign}\mathcal{V} = +1, \quad \operatorname{sign}\mathcal{W} = -1. \tag{4}$$

- One can add a turn, $l$ or $r$, to the route $\mathcal{V}$ from the right. The resulting route is denoted as $\mathcal{V}l$ or $\mathcal{V}r$, respectively. E.g. for routes (3) one has

$$\mathcal{V}l = rlll, \quad \mathcal{V}r = rllr, \quad \mathcal{W}l = lrl, \quad \mathcal{W}r = lrr. \tag{5}$$

- One can remove the last (i.e. the rightmost) turn from the route $\mathcal{V}$. The resulting route is denoted as $\mathcal{V}^{\smile}$. E.g. for routes (3) one obtains

$$\mathcal{V}^{\smile} = rl, \quad \mathcal{W}^{\smile} = l. \tag{6}$$

- One can remove the first (i.e. the leftmost) turn from the route $\mathcal{V}$. The resulting route is denoted as $^{\smile}\mathcal{V}$. E.g. for routes (3) one obtains

$$^{\smile}\mathcal{V} = ll, \quad ^{\smile}\mathcal{W} = r. \tag{7}$$

- $\mathcal{V}_1$ is the first turn of the route. For routes (3)

$$\mathcal{V}_1 = r, \quad \mathcal{W}_1 = l. \tag{8}$$

It will be important in what follows that

$$\frac{\operatorname{sign}(\mathcal{V}^{\smile})}{\operatorname{sign}(\mathcal{V})} \tag{9}$$

equals +1 or -1 whenever the last turn of $\mathcal{V}$ is $r$ or $l$, respectively. Analogously, the ratio $\operatorname{sign}(^{\smile}\mathcal{V})/\operatorname{sign}(\mathcal{V})$ is determined by the first turn of $\mathcal{V}$.

## 2.2 Strings

In the present subsection a central concept of our construction – a string – is introduced. But first we need to settle the enumeration of vertices and links of the Bethe lattice. To this end we choose the following procedure. First we enumerate the vertices of the sublattice $A$ by the index $\mathbf{j}$. The precise way of such enumeration will not be important and thus is not specified. The $\mathbf{j}$'th vertex of the sublattice A is attached to one $X$-link, one $Y$-link and one $Z$-link. We enumerate these three links by the same index $\mathbf{j}$: $\underset{\mathbf{j}}{X}, \underset{\mathbf{j}}{Y}, \underset{\mathbf{j}}{Z}$. Finally, we enumerate the vertex of the sublattice $B$ attached to the link $\underset{\mathbf{j}}{Z}$ by the same index $\mathbf{j}$.[1]

We will also use a general notation $\underset{\mathbf{j}}{Q}$, where $Q$ can assume the values $X$, $Y$ or $Z$.

Each vertex is associated with a two-dimensional local Hilbert space of a spin 1/2. The corresponding Pauli matrices are denoted as $\sigma_{\mathbf{j}A}^{\alpha}$ or $\sigma_{\mathbf{j}B}^{\alpha}$ (with $\alpha = x, y, z$), depending on which sublattice the vertex belongs to.

A *string*

$$\underset{\mathbf{j}}{Q}{}_{\mathcal{W}}^{\mathcal{V}} \tag{10}$$

---

[1] The reader should be alerted that while the link $\underset{\mathbf{j}}{Z}$ connects vertexes of the sublattices $A$ and $B$ carrying the same index $\mathbf{j}$, this is not the case for the links $\underset{\mathbf{j}}{X}$ and $\underset{\mathbf{j}}{Y}$. E.g. the link $\underset{\mathbf{j}}{X}$ in Fig. 1 connects the $\mathbf{j}$'th vertex of the sublattice A to the $\mathbf{j}_2$'th vertex of the sublattice B.

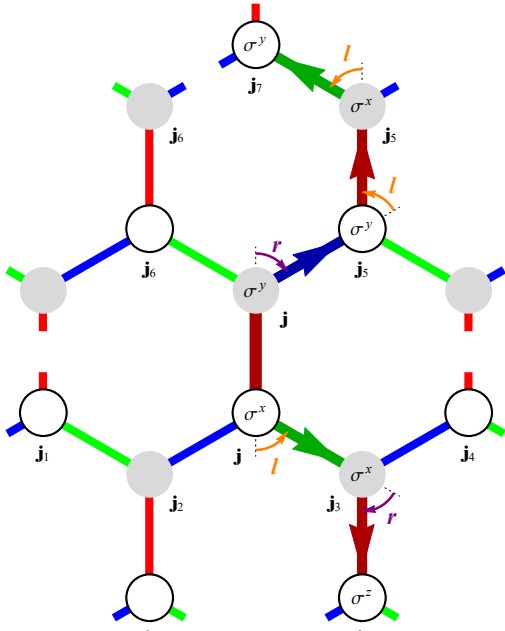

Figure 1: The string $Z_{lr}^{rll} = \sigma_{\mathbf{j}_7 A}^y \sigma_{\mathbf{j}_5 B}^x \sigma_{\mathbf{j}_5 A}^y \sigma_{\mathbf{j} B}^y \sigma_{\mathbf{j} A}^x \sigma_{\mathbf{j}_3 B}^x \sigma_{\mathbf{j}_3 A}^z$ on the Bethe lattice. Vertices of sublattices $A$ and $B$ are shown as open and gray-filled circles, respectively. The $X$-, $Y$- and $Z$-links are shown in blue, green and red, respectively. The links belonging to the string are highlighted.

is a path on the lattice constructed as follows (see Fig. 1 for illustration). Start from the link $Q_{\mathbf{j}}$. Add to it $|\mathcal{W}|$ consecutive links, the first one being attached to the $A$-vertex of $Q_{\mathbf{j}}$. At each step, choose either "left" or "right" turn of the path according to the route $\mathcal{W}$. Then proceed analogously by adding $|\mathcal{V}|$ consecutive links attached to the $B$-vertex of the link $Q_{\mathbf{j}}$. Thus obtained string consists of $|\mathcal{V}| + |\mathcal{W}| + 1$ links.

Each string corresponds to a specific operator, which will be also called "string" and denoted by the same symbol (10). This operator is a product of $|\mathcal{V}| + |\mathcal{W}| + 2$ Pauli matrices residing on the vertices of the string. The choice of a Pauli matrix ($\sigma_x$, $\sigma_y$ or $\sigma_z$) on a specific site is made as follows (see Fig. 1 for illustration). For an end vertex of the string, the Pauli matrix type coincides with the type of the corresponding edge link. For an inner (i.e. non-end) vertex, the Pauli matrix type coincides with that of the link connected to this vertex but not belonging to the string.

Note that a given operator can be represented as a string in multiple ways. For example, for the enumeration of Fig. 1, strings

$$Z_{\emptyset \atop \mathbf{j}_3}^{lrrll}, \quad Y_{rrll \atop \mathbf{j}}^{r}, \quad Z_{lr \atop \mathbf{j}}^{rll}, \quad X_{ll \atop \mathbf{j}_5}^{llr}, \quad Z_{rllr \atop \mathbf{j}_5}^{l}, \quad Y_{\emptyset \atop \mathbf{j}_7}^{rrllr} \tag{11}$$

represent the same operator $\sigma_{\mathbf{j}_7 A}^y \sigma_{\mathbf{j}_5 B}^x \sigma_{\mathbf{j}_5 A}^y \sigma_{\mathbf{j} B}^y \sigma_{\mathbf{j} A}^x \sigma_{\mathbf{j}_3 B}^x \sigma_{\mathbf{j}_3 A}^z$.

Strings of unit length are products of two Pauli matrices residing on the neighbouring vertices, for example $Z_{\emptyset \atop \mathbf{j}}^{\emptyset} = \sigma_{\mathbf{j} A}^z \sigma_{\mathbf{j} B}^z$. They are the building blocks of the Kitaev Hamiltonian (1). The isotropic version of this Hamiltonian, eq. (2), which is studied in the present paper, can be represented as

$$H = -\sum_{\mathbf{j}} \left( X_{\emptyset \atop \mathbf{j}}^{\emptyset} + Y_{\emptyset \atop \mathbf{j}}^{\emptyset} + Z_{\emptyset \atop \mathbf{j}}^{\emptyset} \right). \tag{12}$$

It should be kept in mind that although the three terms $X_{\emptyset_j}^{\emptyset}$, $Y_{\emptyset_j}^{\emptyset}$, $Z_{\emptyset_j}^{\emptyset}$ carry the same index $\mathbf{j}$, they actually belong to different links.

## 3 Deriving Heisenberg equations

### 3.1 Heisenberg representation

We remind that for an arbitrary Schrödinger operator $O$ one can define a Heisenberg operator $O_t$ according to

$$O_t = e^{iHt} O e^{-iHt}. \tag{13}$$

The Heisenberg operator satisfies the Heisenberg equation of motion,

$$\partial_t O_t = i[H, O_t], \tag{14}$$

with the initial condition $O_0 = O$. Given the initial state of the system, $\rho_0$, the corresponding time-dependent expectation value reads

$$\langle O \rangle_t = \operatorname{tr} \rho_0 O_t. \tag{15}$$

Since $[H, O_t] = e^{iHt}[H, O]e^{-iHt}$, taking the time derivative of a Heisenberg operator essentially amounts to commuting the corresponding Schrodinger operator with the Hamiltonian. To avoid further complication of already complex notations, in the next three subsections we will not explicitly introduce the Heisenberg representation, but instead quote the results of commutation of the Schrodinger operators with $iH$.

### 3.2 Time derivative of string operators

An important property of strings is that, as one can easily verify, they form an algebra with respect to commutation. This implies, in particular, that commuting the Hamiltonian (12) with a string always results in a linear combination of strings. These resulting strings can be either shorter or longer by one link than the original string. Explicitly,

$$[iH, Q_{\mathcal{W}_j}^{\mathcal{V}}] = \operatorname{Ex}\left[Q_{\mathcal{W}_j}^{\mathcal{V}}\right] + 2\frac{\operatorname{sign}(\mathcal{V}^{\vee})}{\operatorname{sign}(\mathcal{V})} Q_{\mathcal{W}_j}^{\mathcal{V}^{\vee}} + 2\frac{\operatorname{sign}(\mathcal{W}^{\vee})}{\operatorname{sign}(\mathcal{W})} Q_{\mathcal{W}^{\vee}_j}^{\mathcal{V}}, \qquad |\mathcal{V}|, |\mathcal{W}| \geq 1, \tag{16}$$

$$[iH, Q_{\emptyset_j}^{\emptyset}] = \operatorname{Ex}\left[Q_{\emptyset_j}^{\emptyset}\right], \tag{17}$$

where

$$\operatorname{Ex}\left[Q_{\mathcal{W}_j}^{\mathcal{V}}\right] = 2\left(-Q_{\mathcal{W}_j}^{\mathcal{V}r} + Q_{\mathcal{W}_j}^{\mathcal{V}l} - Q_{\mathcal{W}r_j}^{\mathcal{V}} + Q_{\mathcal{W}l_j}^{\mathcal{V}}\right) \tag{18}$$

is a shorthand notation for expanded strings, and the remaining terms are shortened strings. In the case when one of the routes is empty and another is not we get more complex

formulae:

$$
[iH, \underset{\mathbf{j}}{X_{\emptyset}^{\mathscr{V}}}] = \mathrm{Ex}\left[\underset{\mathbf{j}}{X_{\emptyset}^{\mathscr{V}}}\right] + 2\frac{\mathrm{sign}(\mathscr{V}^{\vee})}{\mathrm{sign}(\mathscr{V})}\underset{\mathbf{j}}{X_{\emptyset}^{\mathscr{V}^{\vee}}} - 2\frac{\mathrm{sign}(^{\vee}\mathscr{V})}{\mathrm{sign}(\mathscr{V})} \times
\begin{cases}
\underset{\mathbf{j}_1}{Y_{^{\vee}\mathscr{V}}^{\emptyset}}, & \mathscr{V}_1 = r, \\[2ex]
\underset{\mathbf{j}_2}{Z_{^{\vee}\mathscr{V}}^{\emptyset}}, & \mathscr{V}_1 = l,
\end{cases}
\tag{19}
$$

$$
[iH, \underset{\mathbf{j}}{Y_{\emptyset}^{\mathscr{V}}}] = \mathrm{Ex}\left[\underset{\mathbf{j}}{Y_{\emptyset}^{\mathscr{V}}}\right] + 2\frac{\mathrm{sign}(\mathscr{V}^{\vee})}{\mathrm{sign}(\mathscr{V})}\underset{\mathbf{j}}{Y_{\emptyset}^{\mathscr{V}^{\vee}}} - 2\frac{\mathrm{sign}(^{\vee}\mathscr{V})}{\mathrm{sign}(\mathscr{V})} \times
\begin{cases}
\underset{\mathbf{j}_3}{Z_{^{\vee}\mathscr{V}}^{\emptyset}}, & \mathscr{V}_1 = r, \\[2ex]
\underset{\mathbf{j}_4}{X_{^{\vee}\mathscr{V}}^{\emptyset}}, & \mathscr{V}_1 = l,
\end{cases}
\tag{20}
$$

$$
[iH, \underset{\mathbf{j}}{Z_{\emptyset}^{\mathscr{V}}}] = \mathrm{Ex}\left[\underset{\mathbf{j}}{Z_{\emptyset}^{\mathscr{V}}}\right] + 2\frac{\mathrm{sign}(\mathscr{V}^{\vee})}{\mathrm{sign}(\mathscr{V})}\underset{\mathbf{j}}{Z_{\emptyset}^{\mathscr{V}^{\vee}}} - 2\frac{\mathrm{sign}(^{\vee}\mathscr{V})}{\mathrm{sign}(\mathscr{V})} \times
\begin{cases}
\underset{\mathbf{j}_5}{X_{^{\vee}\mathscr{V}}^{\emptyset}}, & \mathscr{V}_1 = r, \\[2ex]
\underset{\mathbf{j}_6}{Y_{^{\vee}\mathscr{V}}^{\emptyset}}, & \mathscr{V}_1 = l.
\end{cases}
\tag{21}
$$

Here $\mathbf{j}_1$, $\mathbf{j}_2$, ... $\mathbf{j}_6$ enumerate vertices around the vertex $\mathbf{j}$, as shown in Fig. 1. Analogous formulae are obtained for $\underset{\mathbf{j}}{X_{\mathscr{W}}^{\emptyset}}$, $\underset{\mathbf{j}}{Y_{\mathscr{W}}^{\emptyset}}$, $\underset{\mathbf{j}}{Z_{\mathscr{W}}^{\emptyset}}$. Eqs. (16),(17) and (19)–(21) are straightforwardly obtained by performing commutations. Remind that the ratios of type (9) equal to $\pm 1$, see the explanation below eq. (9). They are used in these equations to account for a sign emanating from the commutations between Pauli matrices.

Let us make a brief remark on deriving analogous equations for Kitaev models on lattices with loops (such as the honeycomb lattice). Direct calculation shows that the equations need to be modified to account for loops. Namely, one needs to modify any equation that involves a string that is one link different from a string with a loop. This is a significant complication, since even simply counting the number of such strings of a given length is a complex problem [38, 39].

### 3.3  Summing over lattice sites

While strings form only a subset of all possible operators, they are still too many to handle. We would like to introduce some sums of strings that can be more tractable. To this end we proceed in two steps. The first one is a straightforward sum over $\mathbf{j}$. In order to obtain well-defined quantities in the thermodynamic limit, we simultaneously perform the normalization:

$$
\sum_{\mathbf{j}}{}' \cdots \equiv \lim_{N \to \infty} (1/N) \sum_{\mathbf{j}} \cdots .
\tag{22}
$$

Here $N$ is the number of sites of one sublattice. This way we define

$$
Q_{\mathscr{W}}^{\mathscr{V}} = \sum_{\mathbf{j}}{}' \underset{\mathbf{j}}{Q_{\mathscr{W}}^{\mathscr{V}}}.
\tag{23}
$$

One can easily see that the commutation of the Hamiltonian with operators $Q_{\mathscr{W}}^{\mathscr{V}}$ are given by eqs. (16)–(21) with all subscripts $\mathbf{j}$, $\mathbf{j}_1$–$\mathbf{j}_6$ removed.

## 3.4 Summing over strings of equal length

The second summation is crucial for obtaining tractable Heisenberg equations. We define

$$Q^{mn} = \frac{1}{2}\left(\frac{1}{\sqrt{2}}\right)^{n+m} \sum_{\substack{\mathcal{V},\mathcal{W}:\\ |\mathcal{V}|=m\\ |\mathcal{W}|=n}} \text{sign}\,\mathcal{V}\,\text{sign}\,\mathcal{W}\left(Q_{\mathcal{W}}^{\mathcal{V}} + Q_{\mathcal{V}}^{\mathcal{W}}\right), \tag{24}$$

where $m, n = 0, 1, 2, \dots$ Obviously, $Q^{mn} = Q^{nm}$.

From eqs. (16),(17) we obtain

$$[iH, Q^{mn}] = -2\sqrt{2}\left(Q^{(m+1)n} + Q^{m(n+1)} - Q^{(m-1)n} - Q^{m(n-1)}\right) \qquad m, n \geq 1, \tag{25}$$

$$[iH, Q^{00}] = -2\sqrt{2}\left(Q^{10} + Q^{01}\right) = -4\sqrt{2}\,Q^{01}. \tag{26}$$

When $m = 0$ but $n \geq 1$, we need to use eqs. (19),(20),(21). This way we obtain

$$[iH, Z^{0n}] = -2\sqrt{2}\left(Z^{1n} + Z^{0(n+1)} - Z^{0(n-1)} + \frac{1}{2}\left(Y^{0(n-1)} + X^{0(n-1)}\right)\right) \tag{27}$$

and analogous equations for $X^{0n}$ and $Y^{0n}$.

## 3.5 Heisenberg equations

From now on, we switch to the Heisenberg representation according to eq. (13). We introduce Heisenberg operators

$$\begin{aligned}
\mathcal{X}_t^{mn} &\equiv X_t^{mn} - \frac{1}{2}\left(Y_t^{mn} + Z_t^{mn}\right),\\
\mathcal{Y}_t^{mn} &\equiv Y_t^{mn} - \frac{1}{2}\left(Z_t^{mn} + X_t^{mn}\right),\\
\mathcal{Z}_t^{mn} &\equiv Z_t^{mn} - \frac{1}{2}\left(X_t^{mn} + Y_t^{mn}\right).
\end{aligned} \tag{28}$$

From eqs. (25), (26), (27) we obtain the following system of Heisenberg equations:

$$\partial_t \mathcal{Q}_t^{mn} = -2\sqrt{2}\left(\mathcal{Q}_t^{(m+1)n} + \mathcal{Q}_t^{m(n+1)} - \mathcal{Q}_t^{(m-1)n} - \mathcal{Q}_t^{m(n-1)}\right), \qquad m, n \geq 1,$$

$$\partial_t \mathcal{Q}_t^{0n} = -2\sqrt{2}\left(\mathcal{Q}_t^{1n} + \mathcal{Q}_t^{0(n+1)} - \frac{3}{2}\,\mathcal{Q}_t^{0(n-1)}\right), \qquad n \geq 1,$$

$$\partial_t \mathcal{Q}_t^{00} = -2\sqrt{2}\left(\mathcal{Q}_t^{10} + \mathcal{Q}_t^{01}\right) = 4\sqrt{2}\,\mathcal{Q}_t^{01}. \tag{29}$$

Here $\mathcal{Q}_t^{mn}$ is a general notation for $\mathcal{X}_t^{mn}$, $\mathcal{Y}_t^{mn}$ and $\mathcal{Z}_t^{mn}$.

Note that $\mathcal{Q}_t^{mn} = \mathcal{Q}_t^{nm}$. For this reason it suffices to find $\mathcal{Q}_t^{mn}$ for $m \leq n$, which is done in the next section.

## 4 Solution of Heisenberg equations

The solution of the system (29) reads

$$\mathcal{Q}_t^{mn} = \sum_{0 \leq \tilde{m} \leq \tilde{n}} \mathbb{G}_{\tilde{m}\tilde{n}}^{mn}(t)\,\mathcal{Q}^{\tilde{m}\tilde{n}}, \quad m \leq n. \tag{30}$$

Here $Q^{\tilde{m}\tilde{n}} = Q^{\tilde{m}\tilde{n}}_{t=0}$ refers to the corresponding Schrödinger operator, and $\mathbb{G}^{mn}_{\tilde{m}\tilde{n}}(t)$ is the propagator given by

$$\mathbb{G}^{mn}_{\tilde{m}\tilde{n}}(t) = \int_0^\pi \int_0^\pi \frac{dp}{\pi} \frac{dq}{\pi} \, e^{-iE(p,q)t} \chi_{\tilde{m}\tilde{n}}(p,q) \, \xi^{mn}(p,q), \tag{31}$$

with

$$E(p,q) = 4\sqrt{2}(\cos p + \cos q), \tag{32}$$

$$\xi^{mn}(p,q) = e^{\frac{i\pi}{2}(m+n)}\left(\left(\sin(mp)\sin(nq) - 2\sin\big((m+1)p\big)\sin\big((n+1)q\big)\right) + \{m \leftrightarrow n\}\right), \tag{33}$$

$$\chi_{\tilde{m}\tilde{n}}(p,q) = -(2 - \delta_{\tilde{m}\tilde{n}})e^{-\frac{i\pi}{2}(\tilde{m}+\tilde{n})}\sum_{l=1}^\infty \frac{1}{2^l}\left(\sin\big((\tilde{m}+l)p\big)\sin\big((\tilde{n}+l)q\big) + \{\tilde{m} \leftrightarrow \tilde{n}\}\right). \tag{34}$$

This is the main general result of the paper.

Let us briefly explain how the solution (30)-(34) has been found. One can verify that, for any fixed $p$ and $q$, the column vector $\|\xi^{mn}(p,q)\|$ is a right eigenvector of the infinite matrix of the system of equations (29), with $E(p,q)$ being the corresponding eigenvalue. Its form has been guessed based on the experience in solving similar problems [40–43]. As a consequence, $e^{-iE(p,q)t}\xi^{mn}(p,q)$ (multiplied by an arbitrary, time-independent operator) is a solution of the system (29) for any $p$ and $q$ (disregarding the initial conditions). Eq. (30) is a linear combination of such solutions with coefficients $\chi_{\tilde{m}\tilde{n}}(p,q)$ chosen to satisfy the initial conditions. Indeed, one can straightforwardly verify that

$$\mathbb{G}^{mn}_{\tilde{m}\tilde{n}}(0) = \delta^{mn}_{\tilde{m}\tilde{n}}, \quad 0 \le m \le n, \quad 0 \le \tilde{m} \le \tilde{n}, \tag{35}$$

where $\delta^{mn}_{\tilde{m}\tilde{n}} = 1$ if $m = \tilde{m}$, $n = \tilde{n}$, and zero otherwise.

Note that the sum in eq. (34) can be calculated explicitly. For general $\tilde{m}$ and $\tilde{n}$ the result is, however, quite bulky; therefore we do not report it here. The result for the particular case $\tilde{m} = \tilde{n} = 0$ is presented in the next section.

## 5 Dynamics of the bond operator for a product initial state

In the present section we focus on the dynamics of a particular observable for a particular type of the initial state.

The initial state we consider is the translation-invariant or staggered-translation-invariant product state

$$\rho_0 = \prod_{\mathbf{j}}\left(\frac{1}{2}(1 + \mathbf{p}\,\boldsymbol{\sigma}_{\mathbf{j}A})\right)\left(\frac{1}{2}(1 + \eta\,\mathbf{p}\,\boldsymbol{\sigma}_{\mathbf{j}B})\right), \tag{36}$$

where $\mathbf{p} = (p_x, p_y, p_z)$ is the polarization vector inside the Bloch sphere (hence, $\mathbf{p}^2 \le 1$), $\eta = 1$ corresponds to the translation-invariant "ferromagnetic" initial state and $\eta = -1$ – to the staggered "antiferromagnetic" initial state with opposite polarizations on the sublattices $A$ and $B$.

The operator we focus on is the *bond operator* $Z^{00}_t$. Due to the (staggered) translation invariance of the initial state, $\langle Z^{00} \rangle_t = \langle \sigma^z_{\mathbf{j}A} \sigma^z_{\mathbf{j}B} \rangle_t$ for arbitrary $\mathbf{j}$.

It is easy to see that for the initial state (36) $\langle X^{00} \rangle = \eta p_x^2$, $\langle Y^{00} \rangle = \eta p_y^2$, $\langle Z^{00} \rangle = \eta p_z^2$, and all other $\langle Q^{mn} \rangle$ vanish. Indeed, whenever $n > 0$, the expectation values of any two terms in eq. (24) with $\mathscr{W}$ differing solely by the last turn cancel one another.

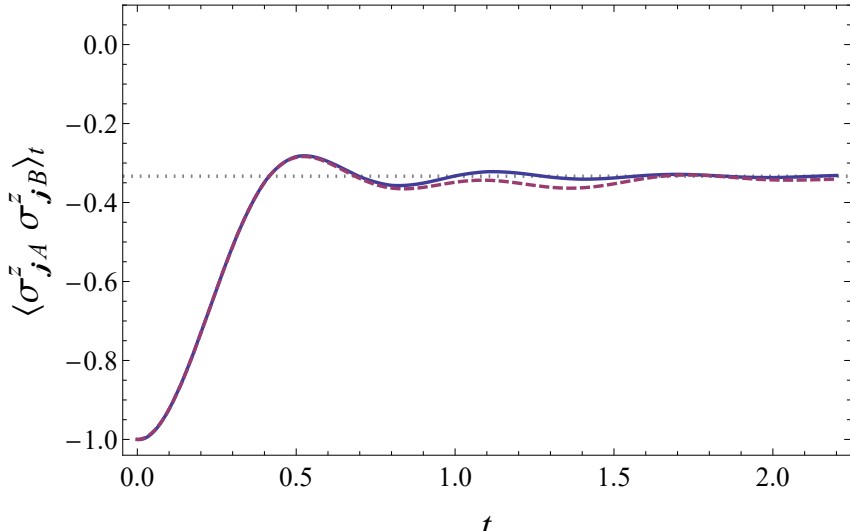

Figure 2: Expectation value of the bond operator for the antiferromagnetic product initial state polarized in the $z$-direction. Solid blue line – the analytical result of the present paper for the Bethe lattice, dashed magenta line – numerical calculations of ref. [34] for the hexagonal lattice. Dotted gray line marks the asymptotic value $(-1/3)$ of the bond operator on the Bethe lattice at $t \to \infty$.

Plugging these expectation values to the general solution (30)-(34) and using the identity $Z_t^{00} = (2/3)\mathcal{Z}_t^{00} - 1/(3N)H$, we finally obtain

$$\langle \sigma_{jA}^z \sigma_{jB}^z \rangle_t = \frac{2}{3}\eta \int_0^\pi \int_0^\pi \frac{dp}{\pi}\frac{dq}{\pi} e^{-iE(p,q)t}\chi_{00}(p,q)\xi^{00}(p,q)\left(p_z^2 - \frac{1}{2}(p_x^2 + p_y^2)\right)$$
$$+ \frac{1}{3}\eta\,(p_x^2 + p_y^2 + p_z^2),\tag{37}$$

where $E(p,q)$ is given by eq. (32), $\xi^{00}(p,q) = -4\sin p \sin q$ according to eq. (33), and $\chi_{00}(p,q)$ can be compactly written as

$$\chi_{00}(p,q) = -12\,\frac{\sin p \sin q}{\left(5 - 4\cos(p-q)\right)\left(5 - 4\cos(p+q)\right)}\,.\tag{38}$$

Eq. (37) is the main result of the present section. In Fig. 2 we plot $\langle \sigma_{jA}^z \sigma_{jB}^z \rangle_t$ for the antiferromagnetic initial condition with $\eta = -1$, $p_z = 1$, $p_x = p_y = 0$.

The asymptotic value of the bond operator at $t \to \infty$ is given by the second term in the right hand side of eq. (37). We note that it would be interesting to compare it to the predictions from the generalized Gibbs ensemble (GGE) [44]. However, we are not aware of any such predictions. Applying the GGE to the Kitaev model might be difficult in practice, since calculating the GGE partition function is expected to be a complex task due to the emergent disorder [45].

For comparison, we also show the evolution of the analogous observable for the same initial condition in the isotropic Kitaev model on the hexagonal lattice, calculated numerically in ref. [34]. One can see that initially the two curves almost coincide, at intermediate times a small discrepancy appear, and at late times this discrepancy almost fades away. The short- and intermediate-time behavior is expectable on physical grounds: at short times the dynamics of the bond operator is determined by the immediate vicinity of the bond, where the two lattices are indistinguishable; the difference in lattice topologies shows up only when the strings of

length six and higher come into play. The closeness of the asymptotic values indicates that the overall contribution of longer strings distinguishing between the two lattices remains limited and quite insignificant.

We would like to emphasize that the formula (37) is equally applicable to arbitrary initial spin polarization $\mathbf{p}$. Amazingly, for $p_z^2 = (p_x^2 + p_y^2)/2$ the expectation value of the $z$-bond operator does not evolve. Furthermore, for $|p_z| = |p_x| = |p_y|$ the expectation values of all three bond operators do not evolve. The latter fact alternatively follows from the symmetry of the Hamiltonian (12) with respect to rotating the spin axes around the $(1/\sqrt{3}, 1/\sqrt{3}, 1/\sqrt{3})$ unit vector by $2\pi/3$ combined with spatial rotation of the lattice by the same angle.

# 6 Summary and outlook

To summarise, we have solved an infinite system of Heisenberg equations for a tailored set of operators in the Kitaev model on the Bethe lattice. These operators are constructed as certain sums of string operators. Wisely composing these sums is crucial for obtaining a tractable system of equations. The solution (30)-(34) of the Heisenberg equations expresses time-dependent Heisenberg operators through time-independent Schrödinger ones. This can be immediately translated to the dynamics of corresponding observables, as soon as the expectation values of the Schrödinger operators in the initial state are given. As an example, we describe the dynamics of the bond operator for a product initial state, see eq. (37) and Fig. 2.

It should be emphasised that we work directly with spin operators and do not resort to the Kitaev's spin-fermion mapping. This way we completely avoid difficulties related to emergent disorder from multiple configurations of IoM values.

Our solution is valid for the isotropic Kitaev Hamiltonian (1),(2). Away from the isotropic point (2), our construction should be refined: while in eq. (24) we sum all strings with $\mathcal{V}$ and $\mathcal{W}$ of given lengths, in the anisotropic case strings with different types of end links should be considered separately. As a result, the set of operators closed with respect to commutation will be much larger, and the Heisenberg equations – more involved. Pursuing this calculation seems feasible, however requires a considerable extra amount of work.

On physical grounds, the dynamics of local observables in the Kitaev model on the Bethe lattice should be close to that on the honeycomb lattice. This intuition is confirmed by comparing our results to the numerical results of ref. [34], see Fig. 2. Extending our method to the honeycomb lattice may prove nontrivial: in contrast to the Bethe lattice, some paths on the honeycomb lattice form closed loops, and this should be accounted for. The latter task can be challenging, since even simply counting the number of self-avoiding paths of a given length on the honeycomb lattice is a complex problem at the frontier of modern mathematics [38, 39].

# Acknowledgements

We are grateful to Louk Rademaker for providing the raw data from his work [34].

**Funding information** O.G. acknowledges support from the Polish National Agency for Academic Exchange (NAWA) through the Grant No. PPN/ULM/2020/1/00247. The work of OL was funded by Russian Federation represented by the Ministry of Science and Higher Education (grant number 075-15-2020-788).

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
