# Peer review of "Out-of-equilibrium dynamics of the Kitaev model on the Bethe lattice via coupled Heisenberg equations"

_SciPost Physics, doi:SciPost Phys. 12, 175 (2022)_

## Round 2 · Referee Report · Benjamin Doyon · 2021-12-17

Strengths
Clear and to the point
Interesting technical idea, showing that it works (proof of concept)
Potential for many new results of interest in quantum many-body
Weaknesses
very special model, still need to see how general the technique can be
Report
In this paper, the authors provide exact solutions to the Heisenberg equations for a particular subset of operators in the Kitaev model on the Bethe lattice. The purpose is to illustrate that it is possible to evaluate exactly time evolution of nontrivial and interesting operators from nontrivial states in spin models, without resorting to exact diagonalisation, and, for instance in the case of the Kitaev model, to mapping to free fermions. This is a short paper, to the point, presenting a clear calculation with explicit results. I find it very well written, and the idea is extremely interesting. The construction is quite stunning, and this, along with previous works by the authors, seems to open up many possibilities, going beyond the standard solution methods. The model here is relatively simple, and the lattice is a regular tree, which drastically simplifies the calculation. Nevertheless, I think the results, and especially the techniques introduces, are extremely interesting.
I do not have any specific comment for improvement, I believe this is publishable as is.

---

## Round 2 · Referee Report · Anonymous · 2021-12-19

Strengths
- innovative idea to access time evolution of kitaev model and in general of a number of interesting many-body system
Weaknesses
- clarifications and some extra checks needed
Report
The idea of using Heisenberg time evolution in many-body systems is usually overlooked, and this paper is a great example of how instead it can be a very useful tool. I recommend publication provided the following points are carefully addressed:
- include a broader introduction of the model, write the Hamiltonian in Pauli matrices and clarify why the Bethe lattice is easier to access with this method compared to hexagonal and other lattices.
- compare the final result with exact results available for quenches with initial states where single fermionic sector can be used
- clarify the meaning of operator of eq 16 and write an explicit derivation of eq 14-19, possible in appendices.
- the large time limit of eq 35 can be reduced to a single integral, reminding of diagonal ensemble. Is there a GGE associated to the large time value of correlator?
Requested changes
see report.

---

## Round 4 · Referee Report · Anonymous (Referee 2) · 2022-4-12

Report

The paper is now ready for publication.

---

## Round 4 · Referee Report · Benjamin Doyon (Referee 1) · 2022-4-19

Report

The authors have replied to the comments made in a satisfactory fashion I believe; in particular the new introduction is indeed helpful. I am happy with accepting this paper at this point.

---

## Round 4 · Author Response

We thank the Referees for the positive evaluation of our work and apologize for the delayed response. Below we address remarks by the second Referee.

Remark: Include a broader introduction of the model, write the Hamiltonian in Pauli matrices and clarify why the Bethe lattice is easier to access with this method compared to hexagonal and other lattices.

Response: We have extended the Introduction according to this advice. The Hamiltonian in terms of Pauli matrices is now given in eq. (1), see also eq. (2). We have also stated in the Introduction that the important feature of the Bethe lattice is the absence of loops, in contrast e.g. to the honeycomb lattice. We have also added a paragraph in the end of Section 3.2 where we elaborate on why loops complicate the analysis.

Remark: Compare the final result with exact results available for quenches with initial states where single fermionic sector can be used

Response: Unfortunately, we are not aware of a prior analytical work that can be compared to our results. In particular, the settings of refs. [29-33] (that are mentioned in the Introduction) are different in the following ways:
- the observables studied (defect density [29-31], Loschmidt echo [32], work statistics [33]) do not belong to the set of observables our method is applicable to;
- the Hamiltonian is changed slowly in time [29,30,33], while our results are for time-independent Hamiltonian;
- the Kitaev Hamiltonian is anisotropic before and after the quench (with an eye on the phase boundary crossing) [29-31,33], while our results are for the isotropic post-quench Hamiltonian.

Remark: Clarify the meaning of operator of eq 16 and write an explicit derivation of eq 14-19, possible in appendices.

Response: Eq. (18) [eq. (16) in the previous version of the manuscript; observe that all equation numbers have been shifted by 2] simply defines a shorthand notation for strings that are longer by one link than the original string. We have clarified this in the text around eq. (18).

Eqs. (16)-(21), while looking bulky, are just the results of a straightforward commutation between the Hamiltonian and a string. One point possibly calling for clarification is how the ratios of type

sign V^/sign V

appear in these equations. In fact, these ratios equal to \pm 1 and are used to account for a sign emerging from the commutations between Pauli matrices, see the text below eq. (9). This explanation has been added below eq. (21).

Remark: The large time limit of eq 35 can be reduced to a single integral, reminding of diagonal ensemble. Is there a GGE associated to the large time value of correlator?

Response: In fact, in the large-time limit the integral in eq. (37) vanishes altogether, and the remaining equilibrium value is given by the constant term in the second line of this equation. This is noted in the paragraph below eq. (38).

It could be indeed interesting to compare the equilibrium values of observables that can be obtained by our method to the predictions of GGE. Unfortunately, we are not aware of any such predictions in the literature. Furthermore, calculating the partition function of the generalized Gibbs state (and thus applying the GGE in practice) is believed to be a complex task, despite the integrability of the model, due to the emergent disorder, see e.g. a discussion in Phys. Rev. B 79, 214440 (2009) (ref. [45]). We have added this remark to the paragraph below eq. (38).

---

## Round 4 · List of Changes

List of changes.

1) Introduction extended, eqs. (1) and (2) added.

2) An explanation of the meaning of the symbol "Ex" is given below eq. (18).

3) Two paragraphs below eq. (21) revised and extended: the appearance of terms of the from (9) explained; complications emerging in lattices with loops highlighted.

4)A remark on the Generalized Gibbs Ensemble added below eq. (38).

5) A number of minor corrections introduced.

---

## Editorial Decision

published